# Silica-Based Stationary Phase with Surface Bound *N*-Acetyl-glucosamine for Hydrophilic Interaction Liquid Chromatography

**DOI:** 10.3390/molecules28207099

**Published:** 2023-10-15

**Authors:** Vaithilingam Rajendiran, Ziad El Rassi

**Affiliations:** Department of Chemistry, Oklahoma State University, Stillwater, OK 74078-3071, USA; vaithilingam.rajendiran@okstate.edu

**Keywords:** *N*-acetylglucosamine-silica, hydrophilic interaction liquid chromatography, derivatized sugars, nucleic acid fragments, benzoic and phenolic acid derivatives

## Abstract

A hydrophilic silica-based stationary phase with surface bound *N*-acetylglucosamine (GlcNAc-silica) was prepared in house and characterized physically via Fourier transform infrared (FTIR) analysis and thermogravimetric analysis (TGA) and chromatographically over a wide range of mobile phase compositions. While both FTIR and TGA confirmed the attachment of the GlcNAc ligands to the silica surface, the chromatographic evaluation of GlcNAc-silica with polar and slightly polar standard solutes (e.g., sugars, nucleic acid fragments, phenolic, and benzoic acid derivatives) yielded the typical hydrophilic interaction liquid chromatography (HILIC) behaviors in the sense that retention increased with increases in solute polarity and the organic content (i.e., acetonitrile) of the hydro-organic mobile phase (i.e., ACN-rich mobile phase). Sugars derivatized with 1-naphthylamine (1-NA) and 2-aminoanthrcene (2-AA) such as xylose, glucose, and short chains maltooligosaccharides constituted the most polar species for HILIC retention evaluation, and in addition, the maltooligosaccharides offered a polar homologous series for gauging the hydrophilicity of GlcNAc-silica in analogy with alkylbenzene homologous series and other nonpolar homologues for evaluating the hydrophobicity of non-polar stationary phases. On the other hand, the benzoic acid and phenolic acid derivatives were the probe solutes for evaluating the HILIC retention dependence of ionizable solutes on the pH of the mobile phase. Similarly, the nucleobase and nucleoside weak basic solutes as well as some typical cyclic nucleotide acidic solutes allowed for the examination of the dependence of solute retention on the pH of the mobile as well as the polarity of the species.

## 1. Introduction

Hydrophilic interaction liquid chromatography (HILIC) is increasingly used for the separation of neutral and charged polar and slightly polar substances on polar stationary phases (i.e., a highly hydrated sorbent). In principle, it exhibits a different selectivity when compared to the traditional reversed-phase chromatography (RPC) and permits the separation of highly polar analytes that are basically un-retained under RPC conditions [1,2,3,4]. Unique selectivity and high separation efficiency are attained in HILIC using polar adsorbents with different structures and nature of functional groups. Stationary phases with amide and hydroxyl functionalities establish the strongest hydrophilicity and retention factors for polar analytes [1,2]. However, novel stationary phases with enhanced efficiency and alternative selectivity are still one of the major directions in the development of the HILIC mode [5,6].

In general, the HILIC separation mechanism is based mainly on the differences in solute partitioning between an adsorbed aqueous layer on the surface of the stationary phase and an organic rich hydro-organic mobile phase. In addition, a variety of hydrophilic interactions including dipole–dipole, ion–dipole, and ion–ion interactions may facilitate the separation of polar solutes on an HILIC column [3,4]. Various polar functional groups such as amino [7], amide [8], diol [9], saccharides [10,11], sulfoalkyl-betaine [12], and cyano [13] groups have been attached to the silica surface to generate HILIC stationary phases with varying degrees of hydrophilic character and separation selectivity. Out of these functional groups, amide, diol, and saccharide bonded stationary phases are neutral (i.e., uncharged) over a wide range of pH used in HILIC separations. Therefore, under these conditions, charged solutes will in principle achieve higher separation efficiency with improved selectivity on neutral rather than charged HILIC stationary phases, due to the absence of electrostatic interactions on the former.

Although many microparticulate silica columns are commercially available, their use in HILIC are marked by some shortcomings associated with the high electrostatic attractions that are exhibited by bare silica as well as poor stability at high pH values, peak tailing, poor reproducibility, low selectivity, and poor resolution due to acidic surface silanol groups interacting with basic compounds in the HILIC mode [3,6]. To circumvent these drawbacks and improve the hydrophilicity, a variety of polar functional groups (e.g., amides, amines, cyano, and diols) have been introduced into the silica support as monomeric or polymeric phases. Based on the surface charge functionalities, the ensuing sorbents were further divided into four types, namely neutral, cationic, anionic, and zwitterionic stationary phases [5].

Some natural biomolecules such as carbohydrates, proteins, peptides, and amino acids, which have multiple polar groups (e.g., amine, carbonyl, polyhydroxy and amide) in their structures, could be attached to silica surfaces to yield stationary phases for achieving HILIC separations [14]. In this category, carbohydrates are broadly used as stationary phases and receive more attention in HILIC separations. The saccharides have more hydroxyl groups and exhibit high hydrophilicity, which makes them more attractive as HILIC stationary phases. In this regard, many solid sorbents have been developed by immobilizing saccharides on the solid support surfaces through “click chemistry” in recent years [15,16]. Some of the HILIC stationary phases with immobilized carbohydrates include maltose [10,17], glucose [14], D-glucamine [2], β-cyclodextrin [18], and chitooligosaccharides [19] just to name a few. In the reported procedures, the sugar molecules were modified with some anchoring groups to facilitate their covalent bonding with the functional groups on the silica surface, in a rather multistep process which makes it very inconvenient to prepare the stationary phases. Moreover, some of these sugar functionalized HILIC stationary phases exhibited strong ionic interactions with ionic analytes because of charged functional groups on the silica surface, leading to poor resolution for ionic compounds. Thus, there is still a need to develop neutral, polar, and robust functionalized silica with sugars.

Though the number of hydroxyl groups on the stationary phase surface is known to influence the hydrophilicity of the stationary phases, only a few literature reports exist on developing HILIC stationary phases with polyhydroxy groups. Most show some limitations such as poor selectivity and retention [20]. Therefore, there is a need to further explore new polyol-based stationary phases to achieve improved HILIC separations. In the current work, a novel acetylamino hydroxylated phase was prepared using *N*-acetylglucosamine. The interest and selection of this sugar were motivated by our previous studies related to amino sugars for HILIC stationary phases, which demonstrated good interactions of polar solutes with organic polymeric monoliths, which were functionalized with amide derivatives of monosaccharides such as D-glucamine to provide good separation selectivity in HILIC [2].

In this study, primarily, the bare silica support was coated with an active epoxy layer via the reaction of silica with γ-glycidoxypropyltrimethoxysilane. The epoxy-coated silica was then functionalized with GlcNAc in the presence of boron trifluoride etherate. The resulting GlcNAc-modified silica provided a convenient and efficient strategy for HILIC separations. In addition to this, the presence of the acetamide group attached using a short alkyl spacer linkage does not have any basic properties. Hence, there are no possibilities for ion-exchange interactions in this situation; ionizable solutes are not retained on the surface via an ion exchange mechanism. By considering the above-mentioned properties, a wide variety of solutes were analyzed, including derivatized mono- and oligosaccharides, benzoic and phenolic acid derivatives, nucleosides and nucleic acid bases, and nucleotides to evaluate the retention characteristics of the GlcNAc-silica stationary phase under investigation in HILIC separations under both isocratic and gradient elution conditions.

## 2. Results and Discussion

### 2.1. Physico-Chemical Characterization of GlcNAc-Silica

As shown in Figure 1, the process of binding of GlcNAc to the silica surface involved two distinct step reactions: the silica was first reacted with epoxy via a reaction with the organosilane γ-glycidoxypropyl trimethoxysilane (GPTMS), which was conveniently attached via the trimethoxysilane groups to the silica surface, which is believed to make a stable anchor, and then followed by a ring reaction opening of the epoxide groups catalyzed by the Lewis acid BF_3_·etherate. Generally, this reaction is performed in an aprotic solvent such as dioxane [21,22,23] but GlcNAc, like other polar sugars, showed poor solubility in the nonpolar dioxane, and therefore, DMF was used instead as a polar aprotic diluent for the epoxide-GlcNAc reaction.

The epoxy coated sorbent with surface-bound GlcNAc was characterized using TGA and FTIR analyses (see Figure 2 and Figure 3). Regarding FTIR, the important feature of the silica spectra is related to the similarities of peaks associated with the silica backbone, such as a sharp band due to siloxane stretching at 1100 cm^−1^ and the nearby band in the region of 810–950 cm^−1^ representing the silanol stretching. However, the strong, broad peak observed in the range of 3220–3530 cm^−1^, in the inset zoomed portion of the spectrum, corresponds to the O-H and NH stretching vibrations, confirming the attachment of GlcNAc moiety onto the silica surface. Furthermore, the stretching vibration peak of C=O and the bending vibration peak of NH around 1685 cm^−1^ represent the characteristic peak of C=O and the secondary amide, thus providing further evidence that the GlcNAc-epoxy silica reaction has been performed successfully.

The TGA curves, which were obtained by heating the silica bound GlcNAc and base silica from 20 °C to 850 °C, are displayed in Figure 3. Around 3.1% weight loss for GlcNAc-silica was observed in the temperature range of 30 to 100 °C, which can be ascribed to the removal of physically adsorbed water molecules. The large weight loss observed between 150 and 850 °C can be attributed to the decomposition of residual organosilane moieties (OSM) and the OSM-GlcNAc ligands as well as to the conversion of silanols to siloxanes. From the starting weight of 9.0 mg GlcNAc-silica, the overall % mass loss was about 8.47%. In this overall %mass loss, about 2% occurred at temperatures ranging from 310 to 340 °C.

### 2.2. Chromatographic Behavior of GlcNAc-Silica Column

Similar to other LC modes, the solutes’ retention on a given HILIC column is controlled by both the composition of the mobile phase and the nature of the retentive stationary phase ligands. Usually, hydro-organic mobile phases are used in HILIC separations, and particularly, an ACN-rich hydro-organic mobile phase is commonly used because of its strong miscible property with water in all proportions. The retention times of polar and slightly polar solutes increase when the %ACN is increased in the hydro-organic mobile phase due to a decrease in its relative polarity making it a weaker solvent. This, in turn, reduces the mobile phase elution power leading to increased solute retention with sharp symmetric peaks. Hence, in order to study the effect of the %ACN on the GlcNAc-bonded-silica stationary phase towards the retention behavior of some typical polar analytes, namely cytidine, adenosine and uridine, plots for the retention factor (k) versus %ACN were constructed, as shown in Figure 4. As can be seen in this figure, and as expected, k of the polar nucleoside solutes separated on the GlcNAc-silica stationary phase increased with increasing %ACN in the hydro-organic mobile phase, showing the typical HILIC retention behavior.

The mobile phase pH and, in turn, solute ionization play certain roles in HILIC separations of charged polar and slightly polar solutes. In fact, the above nucleosides, namely, uridine, adenosine, and cytidine, showed varying k values at two different pH conditions. The k values increased by factors of 1.35 (0.77 vs. 0.57), 1.20 (1.44 vs. 1.21), and 1.10 (2.85 vs. 2.64) for uridine, adenosine, and cytidine, respectively, when going from pH 3 to pH 7 in a mobile phase composed of 97% (*v*/*v*) ACN and 3% (*v*/*v*) 25 Mm ammonium acetate. With the exception of the neutral uridine, the two other nucleosides are weakly protonated at pH 3 and deprotonated at pH 7.0. The increase in the k values may be attributed to increased hydrogen bonding of the nucleosides with the water layer at the surface of the GlcNAc-silica at pH 7.

### 2.3. Retention Behavior of Polar and Slightly Polar Solutes on the GlcNAc-Silica Column

#### 2.3.1. Neutral Polar Solutes: Case of Pre-Column Derivatized Sugars

HILIC is an ideal LC mode for separating inherently hydrophilic molecules such as carbohydrates. In this study, the GlcNAc-silica column was evaluated with 1-NA- and 2-AA-sugar derivatives of xylose, glucose, and maltooligosaccharides under unbuffered mobile phase conditions. As shown in Figure 5, seven solutes of 1-NA- and 2-AA-sugar derivatives were separated on the GlcNAc-silica column. As expected, both sugar derivatives exhibited longer retention times as the %ACN in the hydro-organic mobile phase was increased, a trend that corroborates the HILIC systems of polar solutes reported by others [24]. The 2-AA-sugar derivatives (Figure 5B) yielded peaks not as sharp as those obtained for the 1-NA-sugar derivatives (Figure 5A), which may be due to the stronger aromaticity and hydrophobicity of the 2-AA tag when compared to the 1-NA tag. In all cases, and as expected, smaller derivatized sugar molecules eluted quicker than the larger sugars, since the larger sugars are more polar (i.e., having more hydroxyl groups) than the smaller ones.

A mixture of 1-NA-glucose and maltooligosaccharide derivatives were also separated on the GlcNAc-silica column (see Figure 6). The logarithm of the retention factor (log k values) of the separated glucose and maltooligosaccharides linearly increased with the number of glucosyl units (n_Glc_) in the molecule, confirming the typical HILIC behavior of the GlcNAc-silica column, as shown in Figure 5A. This behavior agrees with the retention model represented by the equation
$$\log\;k=(\log\;\alpha )n_{\mathrm{Glc}}+\log\;\beta$$
where α represents the glucosyl group selectivity increment of the homologous and β represents the k value of their UV tag. The homologous series of maltooligosaccharides constitutes an ideal series for gauging the hydrophilic character of a given HILIC stationary phase. Other linear homo-oligosaccharides (such as chitooligosaccharides) could be used for such evaluations. It should be noted that the use of such a polar homologous series to gauge the hydrophilicity of a given polar stationary phase (e.g., GlcNAc-silica) is analogous to using a nonpolar homologous series (e.g., alkyl benzenes) to gauge the nonpolar character of a given hydrophobic stationary phase (e.g., hydrocarbonaceous bonded silica stationary phases).

In isocratic mode, the 1-NA-maltooligosaccharide derivatives for more than the maltoheptaose mixture were eluted with longer retention times, hence they were chromatographed using the gradient elution mode as shown in Figure 6B. Using the UV detection wavelength at 254 nm and linear gradient elution, ten distinct peaks were separated and detected. In order to determine the limit of detection (LOD), one of the 1-NA derivatized sugars was selected, namely maltotriose. After successive sample dilutions and not exceeding a signal to noise ratio (S/N) of three, the determined LOD for the derivatized 1-NA-maltotriose was approximately 45 ng/Ml on the GlcNAc-silica column.

The GlcNAc-silica column under investigation was compared with a previously introduced D-Glucamine silica column by our research group [2]. For this comparison, the 1-NA- and 2-AA-sugar derivatives were separated on the D-glucamine-silica column under the conditions mentioned in Figure 7. The results show that all seven 1-NA- and 2-AA-sugar derivatives achieved good separation with less retention time on the GlcNAc-silica column than the D-Glucamine-silica column, comparing Figure 5 to Figure 7. The analysis time was about 8 min on the D-Glucamine column versus less than 5 min on the GlcNAc column. This faster elution on the GlcNAc column is due to the presence of an additional acetyl group, which may reduce the polar interactions between the derivatized sugars and the HILIC stationary phase surface, resulting in the GlcNAc column achieving separation in less retention time than the D-Glucamine column for both 1-NA- and 2-AA-sugar derivatives. It should be stated that both stationary phases (i.e., GlcNAc- and D-Glucamine-silica) exhibited similar thermogravimetric behaviors, reflecting very comparable surface ligand densities.

#### 2.3.2. Weakly Acidic Solutes: Case of Phenolic Acids and Benzoic Acid Derivatives

A mixture of three phenolic acid derivatives, namely, sinapic acid (pKa = 3.41), chlorogenic acid (pKa = 3.33), and caffeic acid (pKa = 4.58), was chromatographed on the GlcNAc-silica column under investigation using a mobile phase consisting of 98% ACN:2% 25 mM ammonium acetate (*v*/*v*), at both pH 3 and pH 7. These solutes, which are weakly acidic, were partially ionized at pH 3.0 and fully ionized at pH 7, a fact that increased their partitioning in the adsorbed water layer on the GlcNAc-silica stationary phase as the pH is increased from 3 to 7, thus yielding a noticeable increase in the solute retention time and in turn k values. In other words, the phenolic acid derivatives in the deprotonated state were more polar, which caused strong partitioning into the adsorbed water layer on the HILIC stationary phase. In fact, the k values were 1.30 (pH 7.0) vs. 0.30 (pH 3.0) for sinapic acid, 2.40 (pH 7.0) vs. 0.50 (pH 3.0) for caffeic acid, and 6.50 (pH 7.0) vs. 1.00 (pH 3.0) for chlorogenic acid. It should be noted that the k values increased with increasing the number of hydroxyl groups in the phenolic acids: one (1) –OH for sinapic acid to two –OH for caffeic acid and to four –OH for chlorogenic acid, which follows the typical behavior of HILIC.

Using the same chromatographic elution conditions as those for phenolic acids, three benzoic acid derivatives consisting of *p*-hydroxybenzoic acid (*p*-HBA, pKa = 4.4), gallic acid (pKa = 3.9), and protocatechuic acid (pKa = 4.2) achieved good separation, as shown in Figure 8. It was observed that the behavior of benzoic acids was similar to the behavior of the phenolic acid derivatives; that is, the retention increased with the number of hydroxyl groups, because of the increase in hydrophilic interactions between analytes and the polar sorbent. In fact, *p*-HBA with one hydroxyl group was less retained than protocatechuic acid with two hydroxyl groups, while gallic acid with three hydroxyl groups was the most retained. Also, it should be noted that the k values of acidic solutes (phenolic acids) increased when the pH of the aqueous phase was increased in the hydro-organic mobile phase. At pH 7.0, the unreacted silanol groups will ionize and exhibit cation exchange behavior causing electrostatic repulsion between the acidic solutes and the stationary phase, which in principle should result in decreased solute retention. Thus, the longer retention of solutes at higher pH is considered as a good indication for the effective coverage of the silanol groups by the GlcNAc coating. The same statement could be formulated in the case of the phenolic acid derivatives. The average efficiency obtained for benzoic acid derivatives at pH 7 was 19,750 plates/m (Figure 8B).

#### 2.3.3. Relatively Strong Acidic Solutes: Case of Cyclic Nucleotides

A mixture of three cyclic nucleotides, namely, c-UMP, c-AMP, and c-GMP, was chromatographed on the GlcNAc-silica column using 85% ACN and 15% 25 mM ammonium acetate (*v*/*v*) at pH 3. These solutes eluted in the order of increasing polarity with k values of 1.40 for c-UMP, 2.23 for c-AMP, and 3.20 for c-GMP. The primary amine groups bearing c-AMP and c-GMP eluted after c-UMP may be due to the presence of hydrogen bonding between c-AMP and c-GMP and the hydrophilic GlcNAc sorbent.

#### 2.3.4. Weakly Basic Solutes: Nucleosides and Nucleic Acid Bases

To further illustrate the HILIC retention mechanism of relatively polar solutes on the GlcNAc-silica stationary phase, the separation of a mixture of seven nucleosides and nucleic acid bases is shown in Figure 9. The average efficiency was 17,000 plates/m using a mobile phase consisting of 97% ACN and 3% 25 mM ammonium acetate (*v*/*v*) at pH 3. Nucleosides are viewed as relatively polar solutes as they are nitrogenous bases covalently bonded to a sugar moiety. Hence, increased retention times under HILIC conditions can be expected for nucleosides relative to their corresponding nucleic acid bases because of the increased polar character of nucleosides versus nucleobases.

The elution order of the four analyzed nucleosides can be related to their pKa values. In mobile phase pH 3, adenosine (pKa = 3.64), cytidine (pKa = 4.22), and guanosine (pKa = 3.6) will be positively charged because of their lower basicity [25]. In contrast, the weaker acid uridine (pKa = 9.7) is neutral in charge at pH 3. Therefore, uridine with its secondary amine group eluted before the nucleosides with primary amine groups, namely adenosine, guanosine, and cytidine. Cytidine, which contains a ribose sugar, is eluted later because of its strong positive charge and polarity relative to the other nucleic acid bases. Thus, the nucleic acid bases and nucleosides eluted in order of increasing polarity.

The retention of nucleobase/nucleoside pairs, namely uracil/uridine and cytosine/cytidine, were in the expected order (i.e., increasing polarity). But this elution order was reversed in the case of the adenine/adenosine pair, whereby adenine was more retained than adenosine. This may reflect the importance of the secondary amino group in adenine in establishing hydrogen bonding with the GlcNAc ligand of the HILIC column under investigation.

### 2.4. Reproducibility

The column reproducibility and chemical stability were investigated using the k values of three model test solutes, namely uridine, adenosine, and cytidine using a mobile phase consisting of 97% ACN and 3% of 25 mM ammonium acetate (*v*/*v*) at pH 3. Intraday, interday, and column-to-column reproducibility were evaluated in terms of the % relative standard deviation (% RSD). The average % RSD were 2.6%, 3.4%, and 2.7% for intraday (*n* = 3), interday (*n* = 3), and column-to-column (*n* = 2), respectively. In addition, both HILIC columns were continuously used for nearly two months without any apparent loss in solute resolution or pressure fluctuations.

## 3. Experimental Methods

### 3.1. Instrumentation and Procedures

All HPLC separations were performed using Waters Alliance 2629 separation module (Milford, MA, USA), connected with an in-line degasser, a quaternary solvent pump, an autosampler, and a thermostated column compartment. A Waters PDA detector (model 2475) was used to record the signals at 254 nm. All the chromatographic separations were carried out by maintaining the column at room temperature. *N*-Acetylglucosamine functionalized-silica (GlcNAc-silica) slurry was packed in stainless-steel columns using a constant pressure pump from Shandon Southern Products Ltd. (Rincon, Cheshire, UK). Data acquisition was performed using Empower 2 (Build 2154) software (Waters Chromatography), and then the offline chromatographic data were processed using OriginPro v8.5.1 (Origin Lab Corp., Northhampton, MA, USA). Fourier transform infrared (FTIR) analyses were carried out for the characterization of GlcNAc-silica sorbent using attenuated total reflectance mode on a Nicolet IS50 FT-IR instrument from Thermo Scientific Co. (Waltham, MA, USA). Furthermore, the functionalization of silica with surface bound *N*-acetylglucosamine was assessed via thermogravimetric analysis using a Q-50 thermogravimetric analyzer from TA instruments (New Castle, DE, USA). Approximately 8–10 mg of the samples was heated from 20 °C to 900 °C at a heating rate of 20 °C per min with a 40 mL/min continuous nitrogen gas flow.

### 3.2. Reagents and Materials

Vydac silica with a 5 µm average particle diameter and 300 Å average pore size was obtained from Grace (Hesperia, CA, USA). *N*-Acetyl-D-glucosamine (GlcNAc) was purchased from CN Biosciences, Inc. (Waukegan, IL, USA). The γ-glycidyloxypropyl trimethoxysilane, gallic acid, sinapic acid, and caffeic acid were purchased from Aldrich Chemical Co. (Milwaukee, WI, USA). Analytical grade sugar standards, namely xylose, glucose, maltotriose, maltotetraose, maltopentaose, maltohexaose, maltoheptaose, adenine, adenosine, uracil, uridine, guanosine, cytidine, cytosine, *p*-hydroxybenzoic acid (pHBA), protocatechuic acid, uridine 2′:3′-cyclic monophosphate (cUMP), adenosine 2′:3′-cyclic monophosphate (cAMP), guanosine 2′:3′-cyclic monophosphate (cGMP), sodium cyanoborohydride, and thiourea were purchased from Sigma Chemical Co. (St. Louis, MO, USA). ACS grade acetonitrile, isopropanol, and toluene were obtained from Pharmco-AAPER (Brookfield, CT, USA). Ammonium acetate and formic acid were obtained from Spectrum Quality Products (New Brunswick, NJ, USA). Acetic acid, *N*,*N*′-dimethylformamide (DMF), and phosphoric acid were obtained from EM Science (Gibbstown, NJ, USA). Analytical grade sodium hydroxide was purchased from EMD Chemicals Inc. (Darmstadt, Germany), while maltose and hydrochloric acid were purchased from Fischer Scientific Co. (Fair Lawn, NJ, USA). 1-Naphthylamine (1-NA) was purchased from AK Scientific (Union City, CA, USA), while 2-amino anthracene (2-AA) was obtained from AmBeed Chemicals Inc. (Arlington, IL, USA).

### 3.3. Preparation of Epoxy Bonded Silica and Functionalization with N-Acetylglucosamine

Epoxy-bonded silica was prepared following a previously established procedure [1,26]. Typically, 2.5 g of dry Vydac silica gel was suspended in 30 mL of dry toluene in a round-bottomed flask, and the mixture was heated to 110 °C while slowly stirring to make a slurry. To this suspension, 2.5 mL of γ-glycidoxypropyltrimethoxysilane was added, and the reaction mixture was allowed to react overnight (approx. 18 h), while stirring and heating at 110 °C occurred. The resulting epoxy-silica was isolated via centrifugation and washed successively using toluene and acetone. The epoxy-silica was dried under the hood, followed by an oven at 60 °C and then used for further modification.

The dry epoxy Vydac silica was transferred to a round-bottomed flask that contained 30 mL of DMF. To this suspension, 50 µL boron trifluoride etherate was added while stirring the suspension, and the reaction was continued with stirring for 2 h in the presence of 0.6 g GlcNAc dissolved in DMF.

### 3.4. Column Packing

A slurry of the functionalized silica at 10% (*w*/*v*) was prepared by dispersing 2.0 g of GlcNAc-silica in 20 mL of isopropanol and then sonicated for 20 min to remove dissolved air and to generate a homogeneous slurry. The isopropanol was used as a displacement solution, and the slurry was then packed into a stainless steel column (10 cm × 4.6 mm i.d.) at 6000–7000 psi pressure for 30 min using a constant pressure pump. Finally, the column was washed with water for 10 min at a flow rate of 1 mL/min, followed by equilibration with the running mobile phase (at 1 mL/min) for 30 min before starting the chromatographic analysis.

### 3.5. Derivatization of Mono- and Oligosaccharides

All sugars were derivatized with 1-NA and 2-AA derivatizing agents via a reductive amination reaction using the previously reported procedure [26]. Standard sugars solutions (20 mM) were prepared by using deionized water. The 1-NA derivatizing agent was prepared in 15% (*v*/*v*) glacial acetic acid, whereas the 2-AA solution was prepared in methanol containing 15% (*v*/*v*) glacial acetic acid and DMF (35:65, *v*/*v*). For all experiments, 12 μL of 40 mM freshly prepared derivatizing agent and 3.0 μL of 1 M NaBH_3_CN in THF were added to 10 µL of 20 mM standard sugar. Prepared mixtures were then incubated at 55 °C for 60 min. The derivatized sugar solutions were diluted by a factor of 400 times with the mobile phase and used directly for analysis or stored in the refrigerator until used. The structures of the derivatizing agents are shown in Figure 10.

## 4. Conclusions

In this study, microparticulate silica with a surface-bound GlcNAc ligand, which forms a polyhydroxylated, neutral polar stationary phase, has been shown to offer unique HILIC selectivity and retention behavior to separate a broad range of polar solutes. In fact, baseline separations were achieved for phenolic acids, benzoic acids, nucleosides, and various derivatized sugar solutes due to the unique selectivity of the GlcNAc-silica under investigation, which compensated for the moderate-to-low column efficiency. In addition, the GlcNAc-silica column yielded rapid separations for 1-NA- and 2-AA-sugar derivatives when compared to the previously developed D-glucamine column.

## Figures and Tables

**Figure 1 molecules-28-07099-f001:**
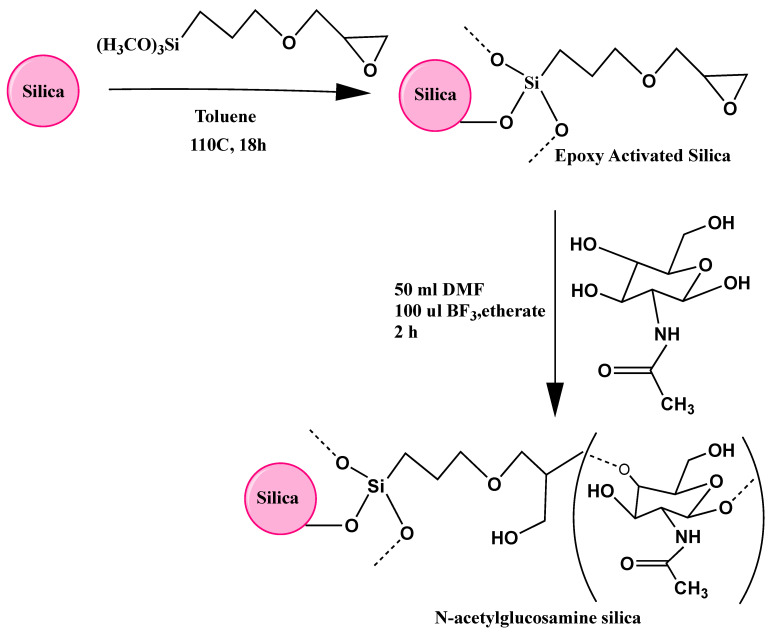
Schematic diagram showing the preparation of the *N*-acetylglucosamine functionalized silica-based stationary phase (GlcNAc-silica). The reaction of the epoxy ring with the hydroxyl groups of the sugar is likely to occur at position 1 or position 4.

**Figure 2 molecules-28-07099-f002:**
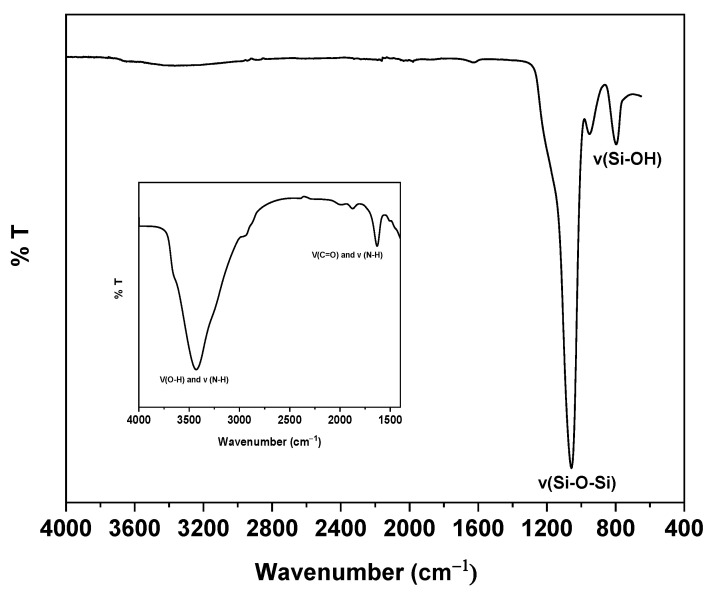
FTIR spectrum of GlcNAc-silica. Inset shows the enlarged region from 1250 cm^−1^ to 4000 cm^−1^.

**Figure 3 molecules-28-07099-f003:**
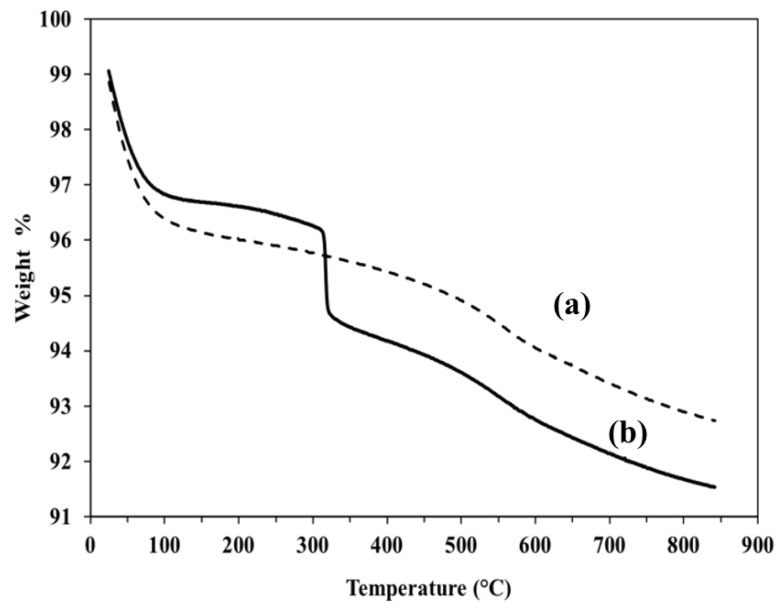
TGA of bare silica in (a) and GlcNAc-silica in (b).

**Figure 4 molecules-28-07099-f004:**
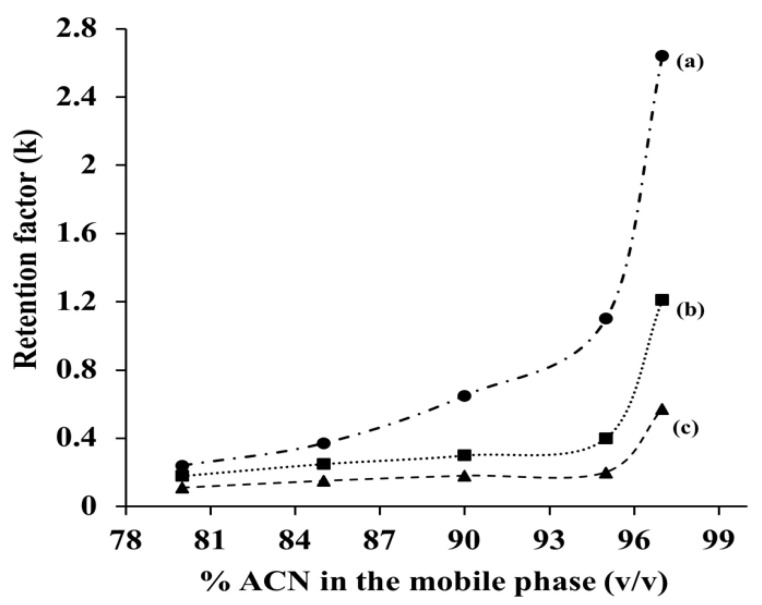
Plot of retention factor (k) versus %ACN (*v*/*v*) in the mobile phase obtained on GlcNAc-silica column. Separation conditions: column dimensions, 10 cm × 4.6 mm id; flow rate, 1 Ml/min; detection, UV at 254 nm; injection volume, 20 µL; column temperature, room temperature; mobile phase, ACN at varying % (*v*/*v*) in 25 Mm ammonium acetate buffer, pH 3. Solutes: (a) cytidine; (b) adenosine; (c) uridine.

**Figure 5 molecules-28-07099-f005:**
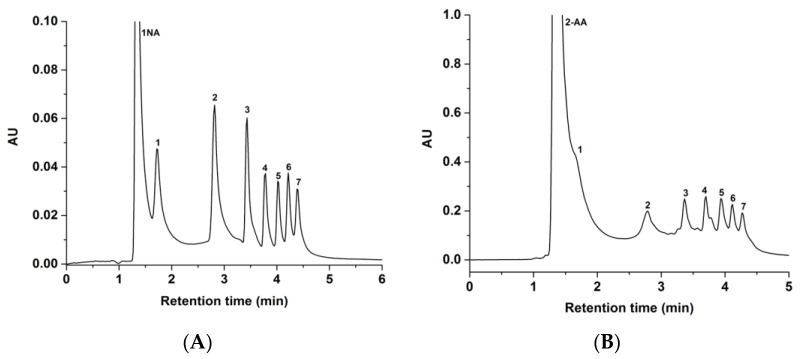
Chromatograms of 1-NA-standard sugar derivatives in (**A**) and 2-AA-standard sugar derivatives in (**B**) obtained on the GlcNAc-silica column. Separation conditions: 4 min linear gradient elution from 95% ACN:5% H_2_O (*v*/*v*) to 70% ACN:30% H_2_O (*v*/*v*) and holding for 2 min isocratic with 70% ACN:30% H_2_O (*v*/*v*); other conditions are the same as in Figure 4. Solutes: 1-NA-sugar or 2-AA-sugar derivatives of 1, xylose; 2, glucose; 3, maltose; 4, maltotriose; 5, maltotetraose; 6, maltopentaose; 7, maltohexaose.

**Figure 6 molecules-28-07099-f006:**
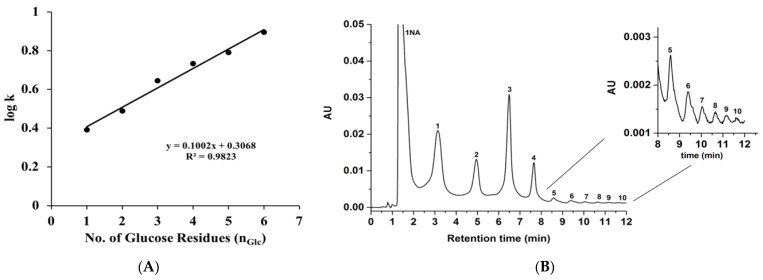
(**A**) plot of log k of 1-NA-derivatized glucose and maltooligosaccharides (up to maltohexaose) separated under isocratic elution on the GlcNAc-silica column using the mobile phase 90% ACN; 10% H_2_O (*v*/*v*). (**B**) Chromatogram of 1-NA derivatized glucose and maltooligosaccharides obtained on the GlcNAc-silica column. Separation conditions: isocratic elution for 1 min with 90% ACN; 10% H_2_O (*v*/*v*) followed by 10 min linear gradient elution from 90% ACN:10% H_2_O (*v*/*v*) to 70% ACN:30% H_2_O (*v*/*v*) and holding for 1 min isocratic with 70% ACN:30% H_2_O (*v*/*v*); other conditions are the same as in Figure 5. Solutes: 1, glucose; 2, maltose; 3, maltotriose; 4, maltotetraose; 5, maltopentaose; 6, maltohexaose; 7; maltoheptaose; 8, maltooctaose; 9, maltononaose; 10, maltodecaose.

**Figure 7 molecules-28-07099-f007:**
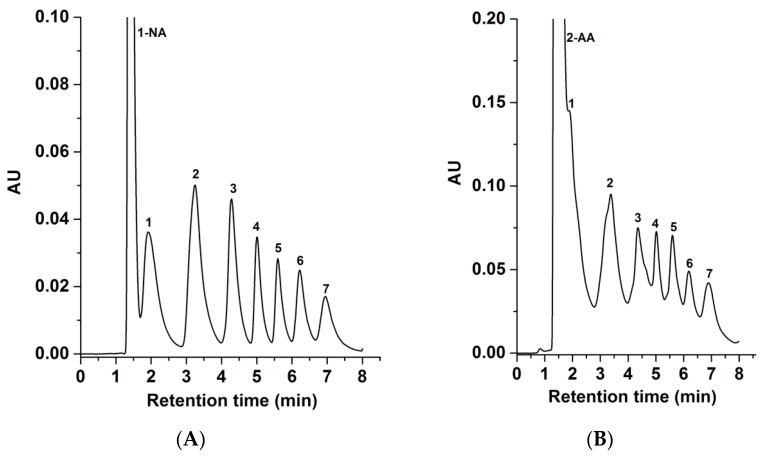
Chromatograms of 1-NA-sugar derivatives in (**A**) and 2-AA-sugar derivatives in (**B**) obtained on the D-glucamine column. Separation conditions: 4 min linear gradient elution from 98% ACN:2% H_2_O (*v*/*v*) to 85% ACN:15% H_2_O (*v*/*v*) and holding for 4 min isocratic with 85% ACN:15% H_2_O (*v*/*v*); other conditions are the same as in Figure 4. Solutes, 1-NA derivatives of 1, xylose; 2, glucose; 3, maltose; 4, maltotriose; 5, maltotetraose; 6, maltopentaose; 7, maltohexaose.

**Figure 8 molecules-28-07099-f008:**
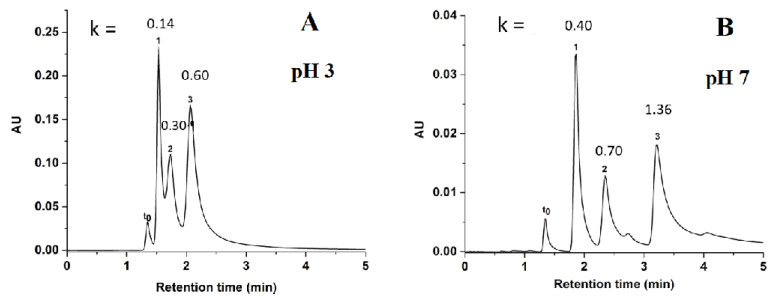
Chromatograms of benzoic acids derivatives including the k values for each peak obtained on GlcNAc-silica column at pH 3.0 in (**A**) and pH 7.0 in (**B**). Separation conditions: isocratic elution with 98% ACN: 2% of NH_4_Ac (*v*/*v*), pH 3.0 in (**A**) or pH 7.0 in (**B**); other conditions are the same as in Figure 4. Solutes: t_0_, toluene; 1, *p*-HBA; 2, protocatechuic acid; 3, gallic acid.

**Figure 9 molecules-28-07099-f009:**
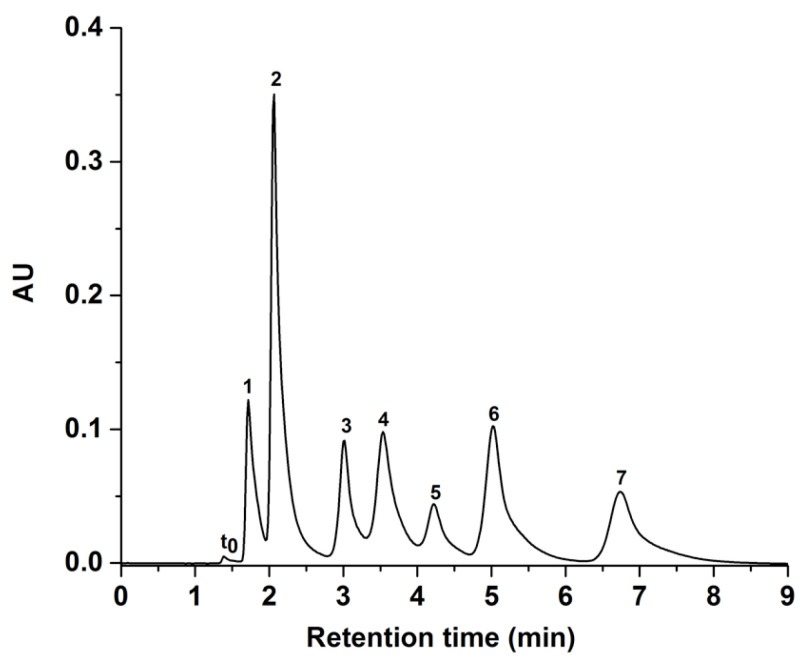
Chromatogram of seven nucleobases and nucleosides obtained on the GlcNAc column. Separation conditions: isocratic elution with 97% ACN:3% of NH_4_Ac (*v*/*v*), pH 3; other conditions are the same as in Figure 4. Solutes: t_0_, toluene; 1, uracil; 2, uridine; 3, adenosine; 4, adenine; 5, cytosine; 6, cytidine; and 7, guanosine.

**Figure 10 molecules-28-07099-f010:**
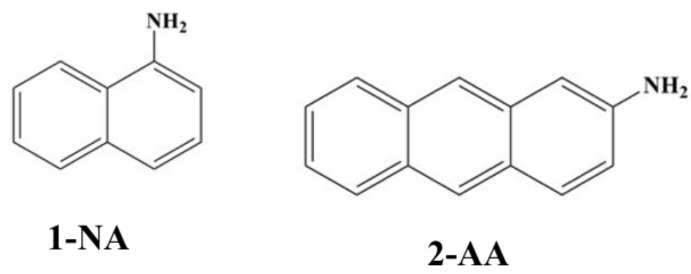
Structures of the derivatizing agents.

## Data Availability

Data are contained within this article.

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
