# Peer review of "Silica-Based Stationary Phase with Surface Bound N-Acetyl-glucosamine for Hydrophilic Interaction Liquid Chromatography"

_molecules, 2023, doi:10.3390/molecules28207099_

Round 1

Reviewer 1 Report

The paper is interesting and well-prepared.  I have, however, some minor comments (my suggestions  for future work, perheps):

1. In my opinion it would be good to present some cost estimate - what I mean is a really rough comparison of the new type of support with commercially available HILIC supports (cost of materials, labor etc).

2.  The stability estimate (Section 3.4) is rather short - did the Authors try to "destroy" the column?  Are there any conditions (mobile phase pH, organic solvents etc) that are NOT recomended in this case?

3.  Did the Authors try to compare the resolution (and order of elution) and the peak shape  of studied compounds on new supports with those on any commercially  available HILIC columns (and on plain silica, maybe)?

Apart from the above suggestions, I would strongly recommend checking the reference list - the Authors cite ref. [30] (line 388), but there are only 25 references listed!

Author Response

For comments 1, 2 and 3 of Reviewer 1, the Reviewer states that "my suggestions  for future work, perheps". For the following comment: Apart from the above suggestions, I would strongly recommend checking the reference list - the Authors cite ref. [30] (line 388), but there are only 25 references listed! We thank the reviewer for spotting ref. [30]: we changed the number to [26] and added the missing reference at the end of "References"

Reviewer 2 Report

Comments on manuscript molecules-2627310, "Silica-Based Stationary Phase with Surface Bound 2 N-acetyl-glucosamine for Hydrophilic Interaction Liquid Chromatography".  In this work, a hydrophilic silica-based stationary phase with bound N-acetylglucosamine was prepared and characterized. The study has been well designed and carried out with extensive experimental work. The article is clearly organized and includes a discussion of the results. I have only a few comments:

1. Line 205 C=C bond perhaps is C=O bond.

2. Lines 214-216 Instead of stating the weight loss over a wide temperature range, the sharp weight loss around 350 °C should be stated,  because this is the most important difference between the proposed stationary phase and the bare silica, 

3. Figure 7. There seems to be an error in Fig 7a, either the figure is incorrect or the figure caption should read chromatogram instead of plot of log k.

4. Some comments on the low column efficiency obtained could improve the article, togheter with future work to improved it.

5. Comparison with a commercial hydroxyl column, pherhaps from bibliography, should have improved the conclusions.

Author Response

Reviewer-2 

The study has been well designed and carried out with extensive experimental work. The article is clearly organized and includes a discussion of the results. I have only a few comments:

  1. Line 205 C=C bond perhaps is C=O bond.

We thank the reviewer for spotting this typo. We changed C=C  to C=O

  1. Lines 214-216 Instead of stating the weight loss over a wide temperature range, the sharp weight loss around 350 °C should be stated, because this is the most important difference between the proposed stationary phase and the bare silica, 

We added after 8.47% the following: In this overall %mass loss about 2% occurred in the temperature ranging from 310 to 340ËšC

  1. Figure 7. There seems to be an error in Fig 7a, either the figure is incorrect or the figure caption should read chromatogram instead of plot of log k.

We thank the reviewer for spotting this error in the manuscript.pdf production. Figure 7 in the manuscript.docx file was replaced by Figure 9 in the manuscript.pdf file, so that figure 9 is used twice in the manuscript.pdf file. Please see the manuscript.docx  

  1. Some comments on the low column efficiency obtained could improve the article, together with future work to improved it.

In the conclusions we added after the phrase “In fact, baseline separations were achieved for phenolic acids, benzoic acids, nucleosides, and various derivatized sugar solutes.” The following: due to the unique selectivity of the GlcNAc-silica under investigation, which  compensated for the moderate to the low column efficiency.

  1. Comparison with a commercial hydroxyl column, perhaps from bibliography, should have improved the conclusions.

It is difficult to compare the data presented by different column manufacturers or between different research groups because the conditions often vary considerably. So, we only compared with previously introduced homemade hydroxyl column (D-glucamine-silica column) by our research group. The results are discussed in section 3.3.1.